# Cabozantinib Is Effective in Melanoma Brain Metastasis Cell Lines and Affects Key Signaling Pathways

**DOI:** 10.3390/ijms222212296

**Published:** 2021-11-14

**Authors:** Trond Are Mannsåker, Tuyen Hoang, Synnøve Nymark Aasen, Ole Vidhammer Bjørnstad, Himalaya Parajuli, Terje Sundstrøm, Frits Alan Thorsen

**Affiliations:** 1Department of Biomedicine, University of Bergen, Jonas Lies vei 91, 5009 Bergen, Norway; trond.mannsaker@student.uib.no (T.A.M.); tuyen.hoang@uib.no (T.H.); synnove.nymark.aasen@hvl.no (S.N.A.); ole.bjornstad@uib.no (O.V.B.); himalaya.parajuli@uib.no (H.P.); 2Department of Neurosurgery, Haukeland University Hospital, Haukelandsveien 22, 5021 Bergen, Norway; terje.sundstrom@uib.no; 3Department of Clinical Medicine, University of Bergen, Jonas Lies vei 87, 5009 Bergen, Norway; 4Molecular Imaging Center, Department of Biomedicine, University of Bergen, Jonas Lies vei 91, 5009 Bergen, Norway

**Keywords:** brain metastasis, melanoma, cabozantinib, apoptosis, PDGF-Rα, IGF-1R, MERTK, DDR1, MAPK pathway, PI3K pathway

## Abstract

Melanomas have a high potential to metastasize to the brain. Recent advances in targeted therapies and immunotherapies have changed the therapeutical landscape of extracranial melanomas. However, few patients with melanoma brain metastasis (MBM) respond effectively to these treatments and new therapeutic strategies are needed. Cabozantinib is a receptor tyrosine kinase (RTK) inhibitor, already approved for the treatment of non-skin-related cancers. The drug targets several of the proteins that are known to be dysregulated in melanomas. The anti-tumor activity of cabozantinib was investigated using three human MBM cell lines. Cabozantinib treatment decreased the viability of all cell lines both when grown in monolayer cultures and as tumor spheroids. The in vitro cell migration was also inhibited and apoptosis was induced by cabozantinib. The phosphorylated RTKs p-PDGF-Rα, p-IGF-1R, p-MERTK and p-DDR1 were found to be downregulated in the p-RTK array of the MBM cells after cabozantinib treatment. Western blot validated these results and showed that cabozantinib treatment inhibited p-Akt and p-MEK 1/2. Further investigations are warranted to elucidate the therapeutic potential of cabozantinib for patients with MBM.

## 1. Introduction

Melanoma incidence rates are steadily increasing and over a third of patients with a metastatic disease have brain metastases at the time of diagnosis [1]. Around 40% of melanomas have a *BRAF* mutation, most frequently *BRAF^V600E^* [2]. BRAF inhibitors such as vemurafenib or dabrafenib have shown intracranial responses in most patients [3]. However, the responses are typically incomplete and short-lived due to intrinsic and extrinsic resistance mechanisms. Immunotherapies have also increased melanoma survival but it remains difficult to predict responses [4]. The drug therapy of brain metastases is further hampered by a limited drug penetration beyond the blood–brain barrier (BBB) [5,6].

Receptor tyrosine kinases (RTK) are cell surface receptors that play important roles in the regulation of gene transcriptions, cell proliferation and cell cycles. Consequently, mutations in RTKs can lead to an altered RTK expression and the development of cancer [7]. Cabozantinib is an oral multi-target RTK inhibitor, which is currently FDA approved for the treatment of renal, thyroid and liver cancers [8]. Growing evidence suggests that cabozantinib may be effective against brain tumors [9,10,11]; recently, the drug demonstrated an intracranial response rate of 55% in brain metastases from renal cell carcinomas [12]. Cabozantinib has repeatedly been shown to inhibit the phosphorylation of RTKs such as c-Met, AXL and VEGFR [13], which are frequently dysregulated in melanomas.

In this study, we report novel findings on the cellular and molecular effects of cabozantinib on melanoma brain metastasis (MBM) in vitro.

## 2. Results

### 2.1. MBM Cell Viability and Colony Formation Is Decreased after Cabozantinib Treatment

To examine if cabozantinib affected MBM cell growth, we conducted monolayer cell viability assays on H1, H3 and H10 cell lines. Morphological changes were observed in the H1 cells, which exhibited a more elongated shape upon cabozantinib treatment. The H3 cell morphology appeared to be unchanged and H10 cells became more rounded after the treatment (Figure 1a). A dose-dependent decrease in the number of viable cells was observed after 72 h of treatment with IC_50_ doses of 71.8 μM, 33.4 μM and 70.4 μM for the H1, H3 and H10 cell lines, respectively (Figure 1b).

Tumorsphere viability assays were carried out to investigate the effects of cabozantinib on colony formation and anchorage-independent growth. After 14 days of treatment, the colonies were both fewer in number and smaller in size compared with the untreated controls (Figure 2a). The IC_50_ values were 4.9 μM, 0.8 μM and 1.7 μM for the H1, H3 and H10 cells, respectively (Figure 2b,c). Thus, the MBM cells grown as tumorspheres were more sensitive to cabozantinib treatment than the monolayer cultures.

### 2.2. Cabozantinib Inhibits MBM Cell Migration

To assess the effect of cabozantinib on tumor cell migration, scratch wound assays were performed. Cabozantinib inhibited the wound closure in all cell lines in a dose-dependent manner (Figure 3a; Appendix A). A quantification of the wound confluency showed that the highest cabozantinib dose resulted in a wound confluency of around 35% for H1 cells and around 15% for H3 and H10 cells (Figure 3b). Hence, the H1 cells were more resistant to the drug.

### 2.3. Cabozantinib Induces Apoptosis in MBM Cell Lines

To further explore the observed inhibitory effects of cabozantinib on the MBM cell viability and colony formation, apoptosis was studied by flow cytometry. Cabozantinib induced apoptosis in H1, H3 and H10 at 50 μM, 15 μM and 40 μM, respectively (Figure 4; Appendix A). Early apoptosis at 72 h in cells treated with the specified drug dose was 20.8% in H1 cells, 14.2% in H3 cells and 18.1% in H10 cells compared with 7.8%, 6.7% and 5.0 % in untreated cells, respectively (Figure 4b). The results were confirmed by caspase-3/7 experiments (Appendix A).

### 2.4. Cabozantinib Targets a Broad Range of p-RTKs in MBM Cells

To elucidate the molecular mechanism behind the observed cellular effects of cabozantinib, we performed a phosphorylated-RTK (p-RTK) array. The results showed that cabozantinib treatment led to a reduced expression of several p-RTKs in H1 cells. p-IGF-1R, p-MERTK and p-DDR1 displayed the most significant reduction whereas the p-PDGF-Rα protein expression was not a significant decrease (Figure 5; Appendix A).

To verify these results, WBs for p-PDGF-Rα, p-IGF-1R, p-MERTK and p-DDR1 were conducted on the lysates of cabozantinib-treated and untreated cells. All the p-RTKs showed a dose-dependent downregulation in their protein expression across all cell lines after cabozantinib treatment (Figure 6). Downregulation was generally most pronounced in H10 cells and least in H1 cells. p-PDGF-Rα was modestly downregulated in H1 and H3 cells whereas p-MERK and p-IGF-1R showed a higher degree of downregulation in all cell lines. The highest reduction in protein expression levels was observed in p-DDR1 in H3 and H10 cells.

### 2.5. MAPK and PI3K-Akt Signaling Pathways in MBM Cells Are Affected by Cabozantinib

To assess the impact of cabozantinib on the critical MAPK and PI3K-Akt signaling pathways, we performed WBs on the key proteins of the pathways. We found that p-MEK 1/2 was inhibited in H1 and H10 cells and seemingly unaffected in H3 cells (Figure 7). The p-Akt protein level was decreased in H1 and H10 cells whereas it had a trend toward an increase in H3 cells after treatment (Figure 7).

## 3. Discussion

Targeted therapy and immunotherapy are key components of contemporary melanoma treatments. These treatments are hampered by resistance problems and have displayed relatively poor effects on brain metastases. The phase III NIBIT-M2 trial demonstrated a median overall survival (OS) of 29.2 months when combining ipilimumab and nivolumab treatment for MBM patients [14]. However, there was only a 41% 4-year survival rate, indicating the need for additional therapeutic approaches for patients with MBM. A phase II trial of cabozantinib treatment in extracranial metastatic melanomas reported a 55% tumor regression and a 9.4 month median OS when of cutaneous, uveal and mucosal origin [15] but another study of only uveal melanomas (UMs) demonstrated no improvement in the progression-free survival relative to the standard care of chemotherapy [16]. The dismal response in UMs could be due to exclusive mutations in G protein-coupled receptors (GPCRs) as the transactivation of PAR2, a GPCR, actively contributed to RTK inhibitor resistance in lung cancer [17]. Here, we report for the first time the inhibitory effects of the multi-target RTK inhibitor cabozantinib in a panel of MBM cell lines in vitro.

The IC_50_ values of the cabozantinib-treated monolayers were comparable with previous reports [18,19,20,21]. Tumorspheres are regarded to be more representative of solid tumor growth in vivo [22,23]. Cabozantinib inhibited the tumorsphere growth for all cell lines (Figure 2). The IC_50_ doses were only 2.4–6.8% of the IC_50_ doses found in the monolayer experiments. This is a common finding [24] and suggests that MBM cells may be more sensitive to treatment when grown in an anchorage-independent microenvironment.

It is well-known that a melanoma has an elevated capacity for migration and invasion compared with other solid tumors and historic autopsy studies have found brain metastases in more than 70% of melanoma patients [25]. In the present study, exposure to cabozantinib showed a dose-dependent inhibition of the MBM cell migration (Figure 3). Drug doses much lower than the IC_50_ doses were able to inhibit the migration, which was similar to the findings in other studies [26]. It is known from the literature that cabozantinib inhibits melanoma migration and invasion in vitro [13]. Taken together, the viability and migration results implied that cabozantinib may inhibit cell migration at low doses even when cell proliferation is not altered.

Resistance to regulated cell death plays a vital role in carcinogenesis including melanomas [27]. The initiation of the apoptotic cascade leads to the activation of the enzyme caspase-3 resulting in DNA breakage and cytomorphological changes [28]. In the current study, cabozantinib induced an early apoptosis in 14–21% of the cells after 72 h of treatment and also led to the upregulation of caspase-3/7 (Figure 4b; Appendix A).

To elucidate how cabozantinib increased apoptosis, we performed a p-RTK array analysis that showed a broad inhibition of the protein expression levels after the treatment (Figure 5). Our studies indicated that downregulated expression levels of p-PDGF-Rα, p-IGF-1R, p-MERTK and p-DDR1 induced apoptosis in the MBM cell lines (Figure 6); however, the contribution of other p-RTKs could not be ruled out. The gene deletion of *PDGF-Ra* has previously been shown to trigger apoptosis [29] and the ligand IGF has been found to be an apoptosis regulator [30]. Interestingly, the phagocytosis of apoptotic cells is mediated by MERTK in macrophages [31] and DDR1 induces apoptosis through the upregulation of pro-apoptotic tumor suppressors [32].

Targeting these four p-RTKs makes sense from a clinical point of view. Melanoma-derived PDGF upregulates the hyaluronic acid production in fibroblasts, which in turn stimulates the melanoma proliferation [33]. It has been indicated that the IGF-1R expression in melanomas is regulated by the surrounding stromal cells as well as the PTEN and BRAF status [34]. A clinical drug combination study on hepatocellular carcinomas is ongoing due to the immunosuppressive effects of cabozantinib on the tumor microenvironment [35]. The MERTK expression is correlated with the melanoma disease progression [36] and the suppression of MERTK is found to inhibit melanoma cell proliferation and migration in vitro [37]. The receptor DDR1 acts as an adhesion molecule in normal melanocytes and contributes to cellular homeostasis [38]. An elevated expression of DDR1 in melanomas is associated with a poor prognosis and the downregulation of DDR1 inhibits the migration, invasion and survival of melanoma cells [39,40]. To the best of our knowledge, it has never been documented that cabozantinib is able to inhibit p-DDR1.

The MAPK and PI3K/Akt signaling pathways are frequently upregulated in an MBM. The activation of these pathways typically occurs through the phosphorylation of RTKs leading to increased intracellular signaling [41,42]. The *BRAF^V600E^* mutation drives the constitutive activation of the MAPK pathway through p-MEK ½, among others [43]. The expression of p-Akt increases with the melanoma invasion and progression and is correlated with poor patient survival [44].

We showed that cabozantinib treatment inhibited the p-MEK 1/2 and p-Akt expression in the *BRAF^V600E^*-mutated cell lines H1 and H10 (Figure 7). This was consistent with the findings from non-BRAF-mutated cancers [18,45]. In the *BRAF^L577F^*-mutated H3 cell line, p-MEK 1/2 and p-Akt did not seem to be affected by cabozantinib. An increased apoptosis was seen after the treatment of all cell lines including H3. Thus, other signaling pathways than PI3K-Akt and MAPK are likely to be involved in the induction of apoptosis after cabozantinib treatment. For example, the WNT pathway can also be aberrated in melanomas [46]. Interestingly, unpublished data from our lab showed that H3 harbored a mutation in the WNT pathway; more specifically, in the *APC* gene. *NRAS* upstream in both the MAPK and PI3K pathways was also mutated in H3. Cross-talk between the WNT and MAPK signaling pathways has been documented due to *APC* loss and *KRAS* mutation in intestinal cancer [47]. Constitutive activation of NRAS may also explain the lack of downregulation of p-MEK 1/2 and p-Akt in H3; however, this was not investigated further in our work. RTKs can interact directly with other RTKs [48,49,50]. For instance, DDR1 overexpression enhances Akt and Erk1/2 activation in response to IGF-1 [51]. Altogether, one could speculate that cross-talk exaggerated the apoptosis in H3 upon cabozantinib treatment independent of the p-MEK 1/2 and p-Akt expression.

In this study on MBM cell lines, we have demonstrated the promising cellular and molecular effects of cabozantinib. Further investigations are warranted to elucidate its therapeutic potential such as studies using selective RTK^−^/RTK^+^ cell lines, combination therapies with current treatment modalities, BBB penetrance studies and in vivo studies.

## 4. Materials and Methods

### 4.1. Cell Lines and Cell Culture

The Regional Ethical Committee (REC) approved the tissue collection and biobank storage of the tumor biopsies as well as the development and use of the cell lines (REC Approvals 2013/720 and 2020/65185). Written informed consent was obtained from all patients. The cell lines were authenticated by short tandem repeat (STR) fingerprinting.

The H1, H3 and H10 cell lines were established in our laboratory from patient biopsies of an MBM. The H1 patient was operated on for a melanoma on the back and while undergoing radiotherapy of the left axilla, the patient was diagnosed and operated on for a brain metastasis. The H3 and H10 patients had only been operated on the thigh and chest for melanomas, respectively. None of the patients had been treated with systemic agents (e.g., chemotherapy, targeted therapy, immunotherapy) or any other treatment directed at their brain metastases. The *BRAF* mutation status of the H1, H3 and H10 cell lines was determined by a massive parallel sequencing of the tumor DNA as previously described [52]. The H1 and H10 cell lines were *BRAF^V600E^*-mutated whereas the H3 cells were *BRAF^L577F^*-mutated.

All cells were grown in Dulbecco’s Modified Eagle Medium (Sigma-Aldrich Inc., St. Louis, MO, USA, cat. #D5671) supplemented with 10% heat-inactivated newborn calf serum (ThermoFischer Scientific, Waltham, MA, USA), 5 μg/mL Plasmocin (Invivogen, Toulouse, France), 2% L-glutamine (BioWhittaker, Verviers, Belgium), penicillin (100 IU/mL) and streptomycin (100 μL/mL) (BioWhittaker). The cells were cultured in a standard tissue incubator at 37 °C with 100% humidity and 5% CO_2_ and were trypsinized when they attained a 75% confluency using 0.25% Trypsin/EDTA (BioWhittaker).

### 4.2. Drug

Cabozantinib (Chemietek, Indianapolis, IN, USA, cat. #CT-XL184) was dissolved in dimethylsulfoxide (Sigma-Aldrich Inc., cat. #D2438) and stock concentrations of 200 mM were stored at −20 °C in aliquots.

### 4.3. Monolayer Cell Viability Assay

The cell viability was studied using an MTS assay (CellTiter 96™ Aqueous One Solution Cell Proliferation Assay, Promega Corporation, Fitchburg, WI, USA, cat. #G358A). Briefly, the cells were seeded at a density of 5 × 10^3^ cells in a 200 mL culture medium per well in 96-well plates (ThermoFischer Scientific, cat. #167008). The day after, the cells were left untreated to serve as a control or treated with cabozantinib (0.01, 0.1, 0.5, 1, 5, 10, 20, 50 and 100 μM) for a period of 72 h. After the treatment, 20 mL of the MTS solution was added into each well and the cells were incubated for 3 h at 37 °C. The absorbance was measured at 492 nm using a plate reader (Multiscan FC Microplate Photometer, ThermoFischer Scientific) with SkanIt software (ThermoFischer Scientific). In each experiment, 6 wells (*n* = 6) were included for every control and drug concentration and triplicate experiments were performed. Graphs were made after a blank subtraction and IC_50_ doses were calculated using GraphPad Prism version 8 (GraphPad Software, Inc., La Jolla, CA, USA). Morphology pictures were taken with a Nikon TE2000 inverted microscope (Nikon Instruments Inc., Melville, NY, USA).

### 4.4. Tumorsphere Cell Viability Assay

The colony formation was assessed as previously described [53]. Briefly, 2000 cells per well (ThermoFischer Scientific, cat. #167008) mixed with soft agar (Sigma-Aldrich Inc., cat. #A9414) was added on top of a base agar (Becton, Dickinson and Company, Frankin Lakes, NJ, USA, cat. #214230). A total of 100 μL cabozantinib at concentrations of 1, 10 and 50 μM for H1; 0.5, 5 and 15 μM for H3; and 0.5, 5 and 40 μM for H10 was further added per well. The control cells were included in the wells without the addition of cabozantinib. The cells were then incubated for 14 days and the culture medium was changed every third day. To assess the colony formation, microscopy pictures were first obtained using a Nikon TE2000 inverted microscope. Thereafter, a resazurin viability assay was carried out. The results were prepared in GraphPad Prism version 8 (GraphPad Software, Inc.) after a blank subtraction. Each experiment was performed in triplicate.

### 4.5. Cell Migration Assay

H1, H3 and H10 cells were seeded in an ImageLock 96-well plate (Essen BioScience Ltd., Hertfordshire, UK cat. #4379) at a density of 3.0 × 10^4^, 2.5 × 10^4^ and 3.8 × 10^4^ cells per well, respectively. An IncuCyte wound-maker tool (Essen BioScience Ltd., Welwyn Garden City, UK) was employed 48 h later to simultaneously create a uniform wound across all wells. All wells were carefully washed with a preheated culture medium to remove the floating cells before adding 200 μL of a culture medium without or with cabozantinib (0.5, 5 and 15 μM for all cell lines) to each well. The cells were placed in an IncuCyte^®^ Live-Cell Imaging System (Essen BioScience Ltd.) and imaging was carried out every 2 h for 72 h using the 10 × objective. The wound closure was analyzed using the IncuCyte^®^ Scratch Wound Cell Migration Software Module (Essen BioScience Ltd., cat. #9600-0012). Each experiment was performed in triplicate.

### 4.6. Apoptosis Assay by Flow Cytometry

Apoptosis was assessed using an AlexaFluor^®^488 Annexin V/dead cell apoptosis kit (ThermoFischer Scientific, cat. #V13245). For all cell lines, 1.0 × 10^5^ cells were seeded in 3 mL of a growth medium per well in a 6-well plate (ThermoFischer Scientific, cat. #140675). After 24 h of incubation, cabozantinib was added to the cells at final concentrations of 1, 10 and 50 μM for H1; 0.5, 5 and 15 μM for H3; and 0.5, 5 and 40 μM for H10 and incubated for 24, 48 or 72 h. The untreated cells were included as the controls. On the day of analysis, the culture medium was transferred to separate tubes (Sarstedt, Nümbrecht, Germany, cat. #62.554.502). The cell monolayers were then washed with 500 µL PBS and the washing solution was transferred into the corresponding tubes. The remaining adherent cells were trypsinized using 0.25% Trypsin/EDTA (BioWhittaker), collected and added to the respective tube. This was followed by washing and centrifugation at 900 rpm for 4 min. The supernatant was discarded and 100 µL of an Annexin V binding buffer containing 5 µL Annexin V and 1 µL propidium iodide (ThermoFischer Scientific, cat. #V13245) was added to each cell pellet and incubated in the dark for 15 min at RT. The cells were analyzed using a flow cytometer (BD Fortessa, BD Bioscience, San Jose, CA, USA). Fluorescence in the FITC-A and PE-A channels was gated to a two-parameter histogram and analyzed using FloJo software (Tree Star Inc., Ashland, OR, USA). The experiment was repeated three times.

### 4.7. Apoptosis Assay by Caspase-3/7 Activity

Apoptosis was further assessed using the IncuCyte Caspase-3/7 Red Reagent for Apoptosis (Essen BioScience Ltd., cat. #4704). For all cell lines, 1.0 × 10^4^ cells in a 100 mL medium were seeded per well in a 96-well plate (ThermoFischer Scientific, cat. #167008) and incubated for 24 h. The culture medium was then removed and 100 mL of a preheated culture medium without or with cabozantinib (0.5, 5 or 15 μM for all cell lines) containing a red apoptosis reagent was instantly added. The plate was incubated in the IncuCyte^®^ ZOOM Live-Cell Analysis System (Essen BioScience Ltd.) 30 min prior to scanning. The images for each well were carried out every 2 h for 72 h using the 10 × objective, collecting one phase-contrast image and one red fluorescent image each time. The IncuCyte^®^ ZOOM Live-Cell Analysis System was used to determine the confluence and red object count. To normalize for different numbers of cells per well after cabozantinib treatment, the data were presented as ratios of the red object count to the confluence. The experiment was conducted in duplicate (*n* = 4 per experiment).

### 4.8. Receptor Tyrosine Kinase (RTK) Array

To elucidate the molecular effects of cabozantinib, we employed a Human Phospho-RTK (p-RTK) Array Kit (R&D Systems, Minneapolis, MN, USA, cat. #ARY001B). Briefly, H1 cells were untreated or treated with 50 μM cabozantinib for 48 h. The cells were lysed with a kit lysis buffer containing a cocktail of protease inhibitors and phosphatase inhibitors (Roche, Basel, Switzerland, cat. #4693124001 and cat. #04906837001; Tocris Bioscience, Bristol, UK, cat. #1190). The lysates were centrifuged at 14,000 g for 5 min to remove the cellular debris and quantified using a bicinchoninic acid (BCA) protein assay (ThermoFischer Scientific, cat. #23225). The kit arrays were blocked for 1 h before the buffer was aspirated. To each array, 300 μg total protein of a cell lysate diluted in a final volume of 1.5 mL of a blocking buffer was added and incubated overnight at 4 °C. The arrays were washed three times with a washing buffer. An anti-phospho-tyrosine HRP detection antibody was diluted with the accompanying array buffer and pipetted into the two wells with the arrays. The arrays were incubated for 2 h at RT and washed twice. To develop the protein expression levels, an enhanced chemiluminescence kit (ThermoFischer Scientific, cat. #A43841) was used. The membranes were imaged with a 1–10 min exposure time using the LAS3000 imaging system (FujiFilm, Saitama, Japan). ImageJ software version 2.0.0 (National Institutes of Health, Bethesda, MD, USA) quantified the relative expressional levels normalized against the reference spots and presented them as a ratio against a negative control. The experiment was performed in triplicate.

### 4.9. Western Blots

H1, H3 and H10 cells were untreated controls or treated with cabozantinib at 1, 10 and 50 μM (H1); or 0.5, 5 and 15 μM (H3); or 0.5, 5 and 40 μM (H10) for 72 h. The cells were lysed using an ice-cold radioimmunoprecipitation assay (RIPA) lysis buffer (ThermoFischer Scientific, cat. #09901) and supplemented with a cocktail of protease inhibitors (Roche, cat. #4693124001) and phosphatase inhibitors (Roche, cat. #04906837001). The cell lysates were centrifuged at 13,000 rpm for 5 min at 4 °C and the resulting supernatants were used. The total protein concentration was quantified by a BCA protein assay (ThermoFischer Scientific, cat. #23225) and 20–35 μg total protein of each cell lysate was resolved by SDS-PAGE. The separated proteins on gels were transferred to nitrocellulose membranes (GE Healthcare Life Sciences, Chicago, IL, USA, cat. #10600001), which were subsequently blocked in Tris-buffered saline (TBS) containing 0.1% Tween and 5% skim milk (blocking buffer) at RT. After washing twice in 0.1% Tween containing TBS (TBS-Tween), the membranes were incubated overnight at 4 °C in a blocking buffer or an antibody diluent (ThermoFischer Scientific, cat. #00-3218) containing the following antibodies: phospho-PDGFR-α (Cell Signaling Technology, Inc., Danvers, MA, USA, cat. #2992); phospho-MERTK (Abcam, Cambridge, UK, cat. #14921); phospho-IGF1 (Abcam, cat. #39398); phospho-DDR1 (Cell Signaling Technology, Inc., cat. #11994); phospho-MEK 1/2 (Cell Signaling Technology, Inc., cat. #9154); phospho-Akt (Cell Signaling Technology, Inc., cat. #4056S); loading control anti-beta actin (Abcam, cat. #ab8227); and loading control GAPDH (Abcam, cat. #9485). The membranes were washed five times with TBS-Tween and incubated for 1 h with a 1:10,000 dilution of a goat anti-rabbit IgG (H+L) cross-adsorbed secondary antibody (Invitrogen, Waltham, MA, USA, cat. #31462) in a blocking buffer. After washing the membranes 5 times, the proteins were detected using an enhanced chemiluminescence kit (ThermoFischer Scientific, cat. #A43841) and an LAS3000 imaging system (FujiFilm). The protein expressional levels were quantified based on the density of the protein bands detected using ImageJ software version 2.0.0 (National Institutes of Health). The relative protein levels were first normalized against the loading control and then calculated and presented as ratios against the untreated controls. The experiments were done in triplicate.

### 4.10. Statistical Analysis

The differences between the treatment groups were assessed using an unpaired two-tailed *t*-test in either Excel version 16.45 (Microsoft) or GraphPad Prism version 8 (GraphPad Software Inc.) A *p*-value < 0.05 was regarded as statistically significant.

## 5. Conclusions

We showed for the first time that cabozantinib decreased the cell viability, colony formation, anchorage-independent cell growth and migration in human MBM cells in vitro. The drug induced apoptosis through the inhibited expression of the p-RTKs p-PDGF-Rα, p-IGF-1R, p-MERTK and p-DDR1. Cabozantinib also inhibited p-Akt and p-MEK 1/2. Additional in vivo studies are warranted to provide data for future translational use of cabozantinib in the treatment of patients with MBM.

## Figures and Tables

**Figure 1 ijms-22-12296-f001:**
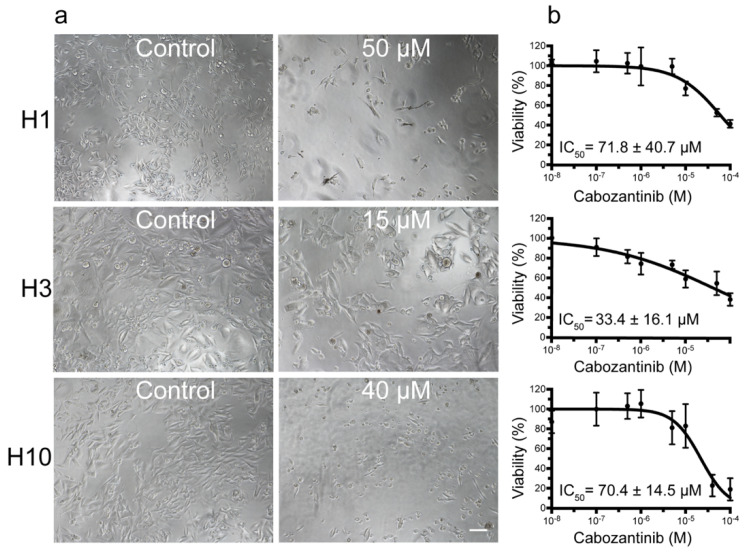
Cabozantinib decreases the viability of melanoma brain metastasis (MBM) cell lines in monolayer cultures. (**a**) Phase-contrast microscopic images (10 × objective) of cells grown as monolayers, either untreated (control) or treated with 50 μM (H1), 15 μM (H3) or 40 μM (H10) cabozantinib for 72 h. Scale bar = 100 μm. (**b**) Representative viability curves of cells grown as monolayers after treatment with cabozantinib with increasing concentrations. The cabozantinib concentrations at which the cell viability was reduced to 50% compared with the control (IC_50_) were calculated from triplicate experiments.

**Figure 2 ijms-22-12296-f002:**
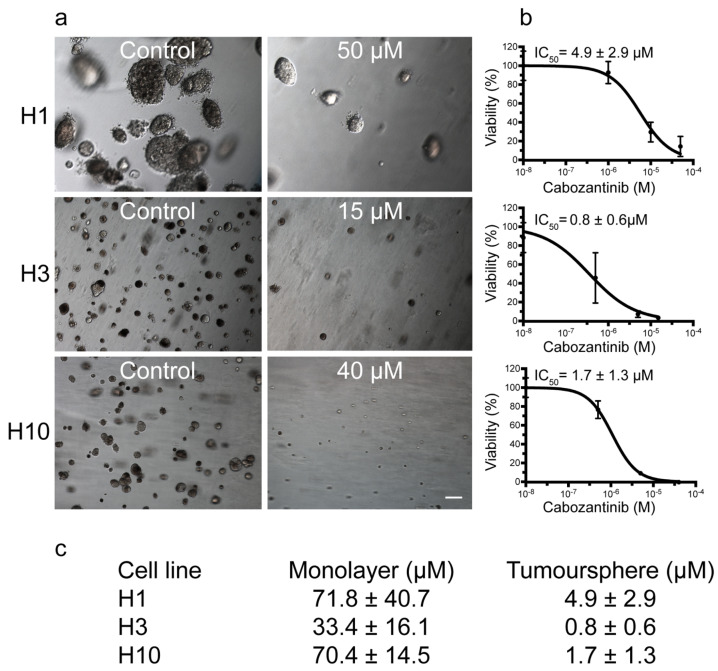
Cabozantinib inhibits colony formation and anchorage-independent growth in MBM cell lines. (**a**) Representative phase-contrast images (10 × objective) at day 14 of H1, H3 and H10 colonies grown in soft agar, either untreated (control) or treated with 50 μM (H1), 15 μM (H3) or 40 μM (H10) cabozantinib. Scale bar = 100 μm. (**b**) Representative viability curves of cells grown as 3D cultures after treatment with cabozantinib with increasing doses. (**c**) Table showing the calculated IC_50_ values of cabozantinib in monolayer cell viability assays and tumorsphere cell viability assays.

**Figure 3 ijms-22-12296-f003:**
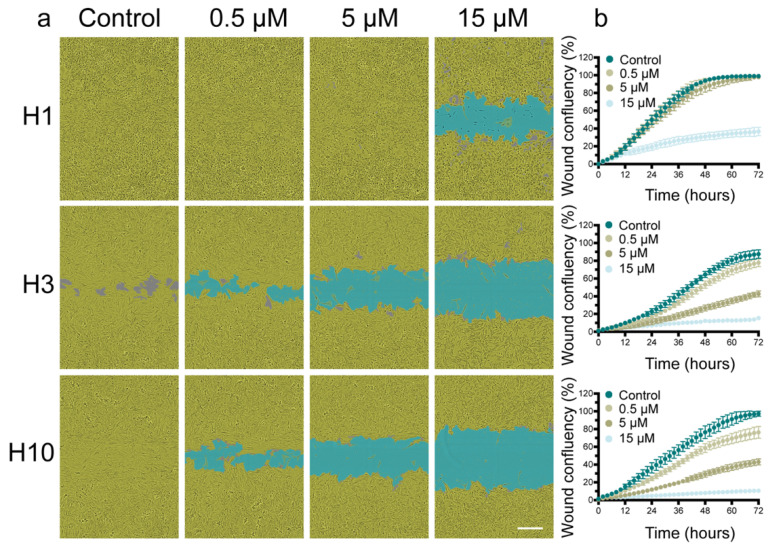
Cabozantinib inhibits the migration of MBM cell lines. (**a**) Representative phase-contrast images (10 × objective) showing the wound confluency obtained 72 h after a scratch wound. Cells were treated with cabozantinib (0.5 μM, 5 μM or 15 μM) or left untreated (control). The MBM cells are colored in yellow and the scratch wound is colored in blue. Scale bar = 300 μm. (**b**) Representative graphs showing the wound confluency over a time period of 72 h.

**Figure 4 ijms-22-12296-f004:**
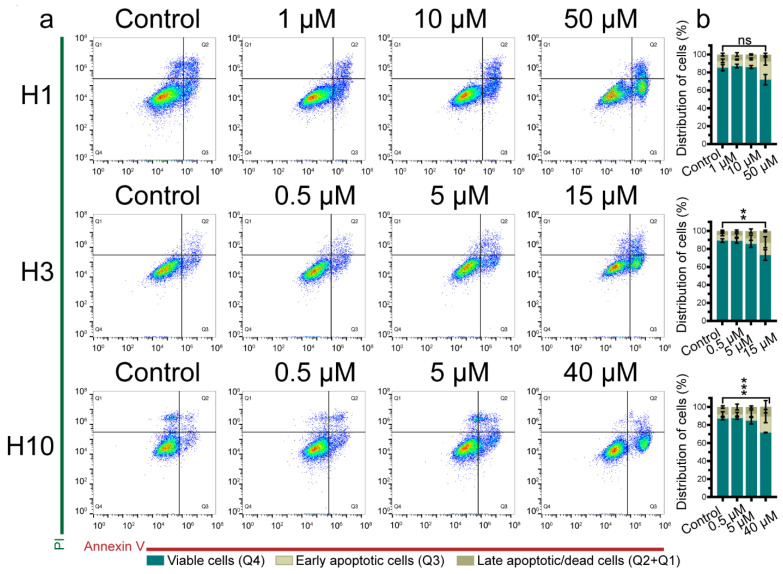
Cabozantinib induces apoptosis in MBM cell lines. (**a**) Representative dot plots of cells untreated (control) or treated with different doses of cabozantinib as indicated for 72 h. Annexin V labels apoptotic cells and propidium iodide (PI) labels necrotic cells. (**b**) Quantification of the percentage of viable, apoptotic and necrotic cells in H1, H3 and H10 cell cultures as untreated (control) or after exposure to cabozantinib. The experiments were performed in triplicate. Q1: necrotic cells; Q2: late apoptotic cells; Q3: early apoptotic cells; Q4: viable cells; ns: not significant; **: *p* < 0.01 in late apoptotic cells between the control and cells treated with the highest drug dose; ***: *p* < 0.001 in viable cells between the control and cells treated with the highest drug dose.

**Figure 5 ijms-22-12296-f005:**
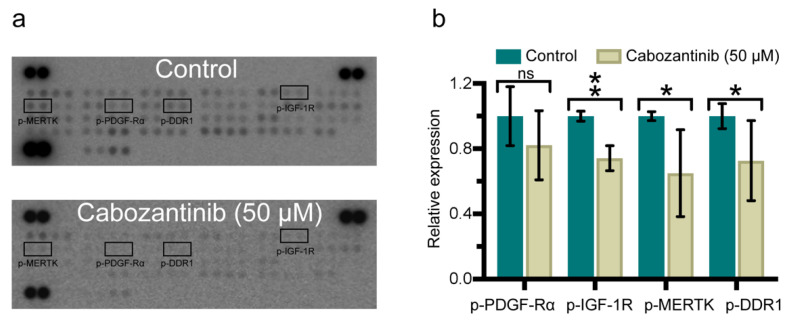
Multiple phosphorylated-receptor tyrosine kinases (p-RTKs) are downregulated in H1 cells after cabozantinib treatment. (**a**) Representative images of RTK arrays that were incubated with lysates of H1 cells untreated (control) or treated with 50 μM cabozantinib for 48 h. Darker spots represent the higher expressional level of the p-RTKs. In each array, each p-RTK is presented by two spots. (**b**) Graphic presentation of the expression levels of selected p-RTKs normalized against the reference spots and presented as a ratio against the controls. ns: not significant; *: *p* < 0.05; **: *p* < 0.01.

**Figure 6 ijms-22-12296-f006:**
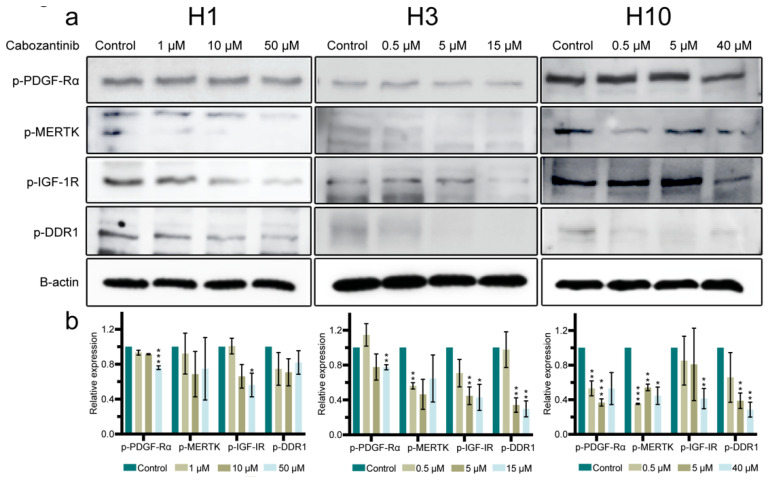
Cabozantinib downregulates the expression of p-PDGF-Rα, p-IGF-1R, p-MERTK and p-DDR1 in MBM cell lines. (**a**) Western blots showing selected p-RTKs in H1, H3 and H10 cells either untreated (control) or treated with 1 μM, 10 μM or 50 μM (H1), 0.5 μM, 5 μM or 15 μM (H3) and 0.5 μM, 5 μM or 40 μM (H10) cabozantinib for 72 h, respectively. (**b**) Quantification of the target proteins detected in the Western blots. The expression levels of the target proteins were normalized against b-actin and compared relatively as a ratio to the control for each specific protein. The Western blot experiments were performed in triplicate. Full-length blots are presented in Appendix A. *: *p* < 0.05; **: *p* < 0.01; ***: *p* < 0.001.

**Figure 7 ijms-22-12296-f007:**
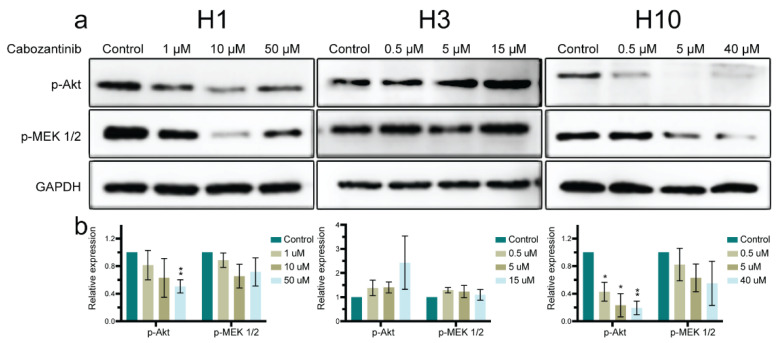
p-Akt and p-MEK 1/2 protein expression levels are downregulated after cabozantinib treatment. (**a**) Western blots showing the expression levels of p-Akt and p-MEK 1/2 after treatment with 1 μM, 10 μM or 50 μM (H1), 0.5 μM, 5 μM or 15 μM (H3) and 0.5 μM, 5 μM or 40 μM (H10) cabozantinib for 72 h, respectively, compared with the untreated (control) cells. The Western blot (WB) experiments were performed in triplicate. (**b**) Quantification of the target proteins detected in the WBs. The expression level of the target proteins was normalized against GAPDH and compared relatively as a ratio to the control for each specific protein. The WB experiments were performed in triplicate. Full-length blots are presented in Appendix A. *: *p* < 0.05; **: *p* < 0.01.

## Data Availability

The datasets used and analyzed during the current study are available from the corresponding author on reasonable request.

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
