# Peer review of "Cabozantinib Is Effective in Melanoma Brain Metastasis Cell Lines and Affects Key Signaling Pathways"

_ijms, 2021, doi:10.3390/ijms222212296_

Round 1
Reviewer 1 Report
Authors report about the anti-tumor activity of cabozantinib in metastatic melanoma to the brain by using three human cell lines from brain metastatic melanoma.
General considerations: The manuscript is clearly written and scientific data are presented in a well-structured manner, so that the experimental design appears appropriate to test the hypothesis and results reproducible based on the details given in the methods section. All the figures/tables/images/schemes are appropriate, informative and easy to interpret, so that the conclusions are consistent with the evidence and arguments presented. However, studies on the possible therapeutic role of Cabozantinib have been performed in vitro and on patients and published over the years with not clearly conclusive results.
Minor criticisism: It is true that brain metastasis are an unmet need in melanoma, but some results have been reached and should be mentioned. For example, I would mention those schemes currently available for the treatment of brain MT from melanoma, like IPI/NIVO combo-immunotherapy (NIBIT M2 trial, Di Giacomo et al. Clin Cancer Res. 2021 Sep 1;27(17):4737-4745) and Targeted Therapies, which have shown some effectiveness in controlling disease over time, expecially when associated to stereotactic radiotherapy. A parallel comparison of Cabozantinib's in vitro efficacy with them, would have been interesting and innovative. Furthermore, those phase II clinical trials published investigating Cabozantinib in uveal and cutaneous melanoma should be mentioned and discussed (include in the references: J Luke et al, Clin Cancer Res, 2020 Feb 15;26(4):804-811). Moreover, this is an in vitro study, and, although experiences are mainly on uveal and cutaneous melanomas, I would have expected this data published before clinical investigations have been performed. Afterwords are quite unuseful, but a confirmation.
Cabozantinib seem to have different effectiveness on uveal or cutaneous metastatic melanoma: could it be related to the differential expression of TKs receptors on cells surface? Could it be possible to know the level of MET expression on brain metastatic cell lines used in the actual study? Is Cabozantinib able to penetrate the blood brain barrier?
Reviewer 2 Report
The authors present detailed evidence by various cell culture and molecular experiments that cabozantinib has a biologic effect on melanoma cells. Despite limited effect on cell viability, cabozantinib seems to display various effects on migration, growth pattern and apoptosis of melanoma cell lines. Especially in melanoma brain metastases, theses effects may be critical.
The authors should include the clinical information, whether the brain metastases which led to cell lines H1, H3 and H10 were resistant or responsive to current or previous treatments at the time of biopsy and which treatments have been performed.
It would be interesting to find the results of the NGS analyses of cell lines H1, H3 and H10 in the supplementary data, if possible.
It is certainly of great interest to further explore (in future studies) the effects of combining cabozantinib with BRAF/MEK inhibitors in cell lines which are BRAF-inhibitor sensitive as well as in cell lines which have acquired secondary resistance, as secondary resistance to BRAF/MEK inhibition is frequent in brain metastases.
